# COLLABORATIVE PROMPT TUNING FOR BLACK-BOX VISION-LANGUAGE MODELS

## ABSTRACT

With the emergence of pretrained vision-language models (VLMs), considerable efforts are devoted to fine-tuning them to downstream tasks. Despite the progress made in designing efficient fine-tuning methods, such methods require access to the model's parameters, which can be challenging due to the high pretraining cost of VLMs. Consequently, model owners often opt to provide the model as a service to safeguard model ownership. In the paper, we propose **C**oll**A**bo**R**ative p**RO**mpt **T**uning (**CARROT**) approach for fine-tuning black-box VLMs to downstream tasks, where we only have access to the input prompts and the output predictions of the model. Specifically, CARROT comprises two modules, a prompt generation module for learning text prompts and a prediction refinement module that enhances output predictions in residual style. Additionally, we introduce an auxiliary prediction-consistent loss to promote consistent optimization across these modules. To optimize the modules, we develop a novel collaborative training algorithm that alternatively optimizes the prompt generation module and the prediction refinement module via the derivative-free and the derivative-based methods, respectively. Extensive experiments on few-shot classification over 15 datasets demonstrate the superiority of CARROT. The results show that CARROT achieves a decent gain of about 12% with 16-shot datasets and only 8,000 queries. Moreover, CARROT trains faster and uses only about 1/80 of the memory footprint for deployment, while sacrificing only 1.62% of performance compared to the white-box method.

## 1 INTRODUCTION

In recent years, large-scale pretrained vision-language models have garnered much attention. By establishing a link between images and natural language, these models exhibit impressive zero-shot capabilities and remarkable transfer ability (Radford et al., 2021; Jia et al., 2021; Alayrac et al., 2022; Li et al., 2022), demonstrating potential in learning open-world concepts. One of the most successful large-scale pretrained vision-language models is CLIP (Radford et al., 2021). By leveraging a massive dataset of 400 million image-text pairs, it learns to align visual and textual representations from a vision encoder and a language encoder, respectively. After pretraining, CLIP (Radford et al., 2021) can perform zero-shot recognition by merely providing the class names. The classification weights are generated by the language encoder through prompting (Liu et al., 2023). For instance, we can adopt a prompt template like "a photo of a {class}" as the input of the text encoder, and then the weights for classification can be synthesized by substituting in the "{class}" with the actual class name. Thus, the classification score is the cosine similarity between the image feature and the weights.

Besides its remarkable zero-shot ability, recent studies have found that CLIP (Radford et al., 2021) also possesses astonishing transfer ability (Zhou et al., 2022b; Zhang et al., 2022; Lu et al., 2022). For example, CoOp (Zhou et al., 2022b) can achieve a 15% improvement compared to zero-shot CLIP (Radford et al., 2021) with only 16 samples per class by fine-tuning a mere 16k parameters. Despite their parameter- and data-efficiency, these methods assume we have access to the model parameters, which is unrealistic in the current era. Training large vision-language models typically requires extensive computational resources and data, thus leading to high training costs. Therefore, model owners seldom release the model and the weights to protect the model ownership. Typically, model owners deploy the models as a service, such as GPT-4 (OpenAI, 2023), where we can only obtain the input and output. Therefore, it is crucial to explore ways to fine-tune powerful vision-language models in the black-box scenario.

To address the aforementioned challenge, we present **CollAboRative pROmpt Tuning** (**CARROT**), a parameter- and data-efficient fine-tuning approach for black-box vision-language models. The CARROT framework comprises three key components. Firstly, it incorporates a prompt generation module designed to learn global text prompts suitable for downstream datasets. Since we do not have access to gradients from the black-box model, we employ derivative-free optimization (DFO) for the module, inspired by prior works (Sun et al., 2022b;a). DFO involves optimizing this module by sampling solutions from a predefined parameterized distribution along with their corresponding loss values. To expedite the optimization process, the text prompts are projected into a lower-dimensional subspace using a random matrix, as Aghajanyan et al. (2021) demonstrates a low-dimensional subspace can be as effective as the full parameter space for fine-tuning.

Secondly, CARROT introduces a prediction refinement module aimed at enhancing output predictions. This module builds upon the predictions of black-box models and is optimized through gradient descent. It consists of a three-layer MLP that learns the prediction's residual, with the residual connection playing a pivotal role in the collaborative training algorithm discussed below.

Thirdly, CARROT develops a novel collaborative training algorithm to optimize the aforementioned modules jointly. Given that the prompt generation module and the prediction refinement module are optimized using different optimizers (derivative-free and derivative-based), their joint training poses a challenge. To address this, we demonstrate that the model with residual connections can be reframed as the addition of outputs of each layer, enabling the modules to be optimized alternately. Fortunately, both VLMs and the prediction refinement module incorporate shortcut connections, facilitating this iterative optimization. To bolster training stability, we introduce a prediction-consistent loss that penalizes deviations between the black-box model's output and the refinement module's output.

Our main contributions are summarized as follows:

- Our method is almost the pioneering work in exploring efficient fine-tuning methods for black-box vision-language models, without requiring access to the models' parameters.

- CARROT provides a new framework for fine-tuning black-box VLMs, incorporating learnable modules into both the input and output of the black-box models.

- CARROT comprises a prompt generation module and a prediction refinement module. These modules are designed to learn the text prompts and refine the output predictions, respectively. In addition, we propose a collaborative training algorithm to train these modules jointly, as they use different optimizers, i.e., derivative-free and derivative-based, and we propose a prediction-consistent loss to enhance training stability.

- CARROT significantly outperforms black-box baselines on 15 datasets on few-shot classification. Compared to the white-box method, CARROT trains faster and requires only 1/80 of the memory footprint for deployment.

## 2 RELATED WORK

**Vision-Language Models.** In recent years, vision-language models (VLMs) have gained popularity as fundamental models that aim to connect the modalities of vision and language. These models are pretrained on large-scale image-text datasets, which endows them with powerful transferable abilities such as zero-shot learning, few-shot adaptation, and in-context learning (Radford et al., 2021; Kim et al., 2021; Lu et al., 2019; Su et al., 2019; Jia et al., 2021). Contrastive-based vision-language pretraining has become the mainstream approach in this field. These methods, including CLIP (Radford et al., 2021) and ALIGN (Jia et al., 2021), are trained on large-scale web-based noisy image-text pairs. They employ a language encoder and a vision encoder to encode the texts and images, respectively, and learn to align their representations through contrastive loss. This loss pulls the representations of matching image-text pairs together and pushes those of mismatched pairs apart.

**Fine-tuning for VLMs.** Inspired by the prior works in NLP, recent researches focus on developing efficient fine-tuning methods for VLMs on downstream tasks (Zhou et al., 2022b;a; Zhang et al., 2022; Gao et al., 2023; Lu et al., 2022; Chen et al., 2023; Derakhshani et al., 2022; Wang et al., 2023). These efficient fine-tuning methods are typically parameter- and data-efficient, requiring only a small number of parameters and utilizing a small set of data, yet achieving a significant improvement over zero-shot learning. Existing efficient fine-tuning methods can be classified into

two categories: prompt tuning (Zhou et al., 2022b;a; Lu et al., 2022; Chen et al., 2023) and adapter-style tuning (Gao et al., 2023; Zhang et al., 2022). Prompt tuning methods propose to learn soft text prompts for downstream tasks through back-propagation on few-shot datasets. For instance, CoOp (Zhou et al., 2022b) proposes to learn global soft text prompts for downstream tasks through back-propagation on few-shot datasets. ProDA (Lu et al., 2022) assumes the output of the prompts follows a normal distribution and attempts to learn a collection of soft text prompts to capture the variational visual representation. Adapter-style tuning methods, on the other hand, maintain the original zero-shot classifier but refine the representation with a lightweight MLP. CLIP-Adapter (Gao et al., 2023) proposed to add a lightweight MLP to refine the visual and text features via the residual connection. Tip-Adapter (Zhang et al., 2022) further improves CLIP-Adapter (Gao et al., 2023) by replacing the MLP with visual prototypes of labeled few-shot data. This not only inherits the training-free advantage of zero-shot CLIP (Radford et al., 2021) but also performs comparably to those training-required approaches. Although these methods have achieved satisfactory results on downstream datasets, they all assume that the entire parameters of VLMs are available, which is unrealistic. Due to the high cost of pretraining large-scale VLMs, the model owners seldom release the model weights to safeguard the model ownership. Therefore, it is necessary to investigate ways to fine-tune black-box vision-language models.

**Black-Box Optimization.** In the field of black-box optimization, two main categories exist: zeroth-order optimization and evolutionary algorithms. Zeroth-order optimization addresses optimization problems similar to gradient-based methods but estimates gradients by sampling instead of back-propagation, which is used in black-box tuning in previous works (Tsai et al., 2020; Oh et al., 2023). In contrast, evolutionary algorithms (Hansen et al., 2003), inspired by biological evolution, generate candidate solutions through a parameterized multivariate normal distribution and update the distribution by selecting a subset of the solutions based on their fitness. BBT (Sun et al., 2022b) and BBTv2 (Sun et al., 2022a) employed CMA-ES (Hansen et al., 2003) to learn text prompts for language models on downstream NLP tasks.

## 3 METHOD

As depicted in Figure 1, we divide the model into three distinct parts: the input space, the black-box vision-language model, and the output space. Since we lack access to the parameters of the black-box model, we can solely integrate learnable modules in the input and output spaces. In the input space, we propose a prompt generation module, which learns global text prompts for downstream tasks using the Covariance Matrix Adaptation Evolution Strategy (CMA-ES) (Hansen et al., 2003). In the output space, we propose a prediction refinement module to refine the output prediction of the black-box model in residual style, which can be optimized using gradient descent. Later, we propose a collaborative training algorithm to train them jointly, despite their utilization of different optimizers.

### 3.1 PROMPT GENERATION MODULE

We consider a black-box vision-language model, denoted by $f$. Given the model, we can only obtain its input and output prediction $f(\{t_k\}_{k=1}^K, \{i_n\}_{n=1}^N) \in \mathbb{R}^{N \times K}$. Here, $\{i_n\}_{n=1}^N$ refers to the $N$ images that are uploaded to the black-box vision-language model, and $\{t_k\}_{k=1}^K$ denotes the $K$ class text prompts, where each prompt $t_k$ consists of a predefined template $p_0$ (e.g., "a photo of a") and a corresponding class name $c_k$. Specifically, we have $t_k = [p_0, c_k], k = 1, 2, \ldots, K$ for $K$ classes.

As shown in Figure 1, we propose to learn global prompts $p \in \mathbb{R}^{n \times d}$ for the black-box model, where $n$ and $d$ represent the length of the prompts and their dimension, respectively. Previous work (Aghajanyan et al., 2021) reveals that a low-dimensional subspace can be as effective as the full parameter space for fine-tuning, we further reduce the search space for fast training by mapping the prompts $p$ into a low-dimensional subspace using a random matrix, i.e., $p = Az$. Here, $A \in \mathbb{R}^{nd \times d_0}$ is a random matrix sampled from a Gaussian distribution, and $d_0 \ll nd$ is the dimension of the subspace. Next, we add the prompts to the initial prompts $p_0$ (e.g., "a photo of a"). Thus, the optimization problem can be formulated as follows:

$$\min_z \mathcal{L}(f(\{[p_0 + Az, c_k]\}_{k=1}^K, \{i_n\}_{n=1}^N), Y), \tag{1}$$

where $\mathcal{L}$ is the cross-entropy loss, and $Y$ denotes the ground-truth labels. Since the model's gradients are not accessible, we solve this problem using a DFO method, CMA-ES (Hansen et al., 2003).

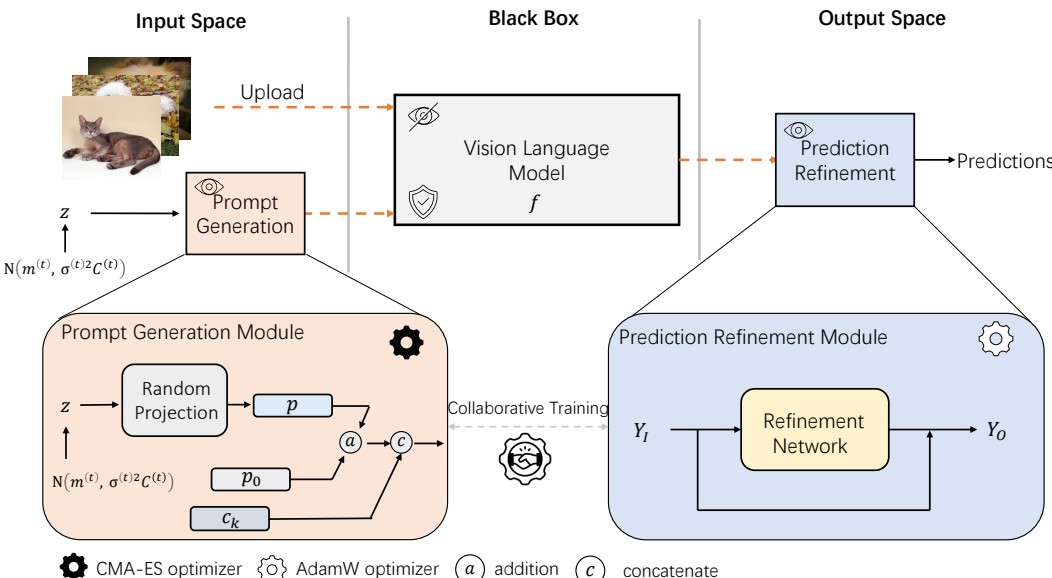

Figure 1: **The architecture of our proposed method.** Our proposed method consists of two modules: a prompt generation module and a prediction refinement module. The prompt generation module utilizes the CMA-ES optimizer to learn the text prompts. Specifically, given $z \in \mathbb{R}^{d_0}$ from the CMA-ES optimizer, we project it into the prompt space using a random matrix $A \in \mathbb{R}^{nd \times d_0}$ and add it to initial prompt embeddings $p_0 \in \mathbb{R}^{n \times d}$ (e.g., "a photo of a"). We then concatenate it with the class name embedding $c_k$ to obtain the final prompts $p_k = [p_0 + Az, c_k]$, where $k = 1, 2, \ldots, K$ for $K$ classes. The prediction refinement module refines the output of the black-box model using a refinement network in residual style. It can be optimized using gradient descent. Since the modules use different optimizers, we propose a novel algorithm to train them collaboratively. For more details on the training process, please refer to section 3.3.

CMA-ES (Hansen et al., 2003) is a parameterized search distribution model that uses a multivariate normal distribution. At each iteration, CMA-ES generates a population of new query solutions by sampling from the multivariate normal distribution model, as follows:

$$z_i \sim m^{(t)} + \sigma^{(t)} \mathcal{N}(0, C^{(t)}). \tag{2}$$

Here, $i$ denotes the index of the sampled solution, $\lambda$ is the population size, $m^{(t)}$ represents the distribution mean, $\sigma^{(t)} \geq 0$ is the step-size, and $C^{(t)}$ denotes the covariance matrix of the distribution. The parameters $m^{(t)}, \sigma^{(t)}, C^{(t)}$ are updated in each iteration to minimize the loss of the sample solutions (cf. CMA-ES (Hansen et al., 2003) for more details).

## 3.2 PREDICTION REFINEMENT MODULE

Besides learning text prompts, we further build a refinement network on top of the output of the black-box vision-language model, which learns to refine the output prediction in residual style.[1] Specifically, given the original output $Y_I \in \mathbb{R}^K$ of the black-box model, the refinement network learns to generate the residual $R(Y_I)$, which is added to the original output to obtain the final result:

$$Y_O = Y_I + R(Y_I). \tag{3}$$

The refinement network is trained to minimize the cross-entropy loss. As the refinement network is built on top of the black-box model and does not require gradients from the model, we can use gradient descent to optimize the refinement network.

---

[1]The residual connection is necessary for collaborative training algorithm 3.3. Table 3 shows its ablation.

### 3.3 COLLABORATIVE TRAINING ALGORITHM

As shown in Figure 1, the prompt generation module and prediction refinement module uses different optimizers (CMA-ES and AdamW, respectively). Therefore, optimizing the modules jointly becomes a challenge. To address this issue, we propose a collaborative training algorithm for them.

Previous works (Mei et al., 2016; Kandasamy et al., 2015; Sun et al., 2022a) have shown that networks with shortcut connections can be decomposed into some additive form. For example, the forward computation of a two-layered model can be decomposed as follows:

$$
\begin{aligned}
f(x_1) &= f_2(x_2) + x_2 \\
&= f_2(x_2) + f_1(x_1) + x_1.
\end{aligned}
\tag{4}
$$

This implies that $f_1$ and $f_2$ can be optimized separately since the function $f$ is decomposed into an additive form. Fortunately, vision-language models typically have residual connections, and the prediction refinement network also comprises a shortcut connection. Therefore, we can iteratively optimize the prompt generation module and the prediction refinement module. Moreover, to enhance training stability, we further propose a prediction-consistent loss. Specifically, we use an additional Kullback–Leibler (KL)

---

**Algorithm 1** Collaborative Training

**Require:** Budget of API calls $\mathcal{B}$,
  Population size $\lambda$,
  Dataset size $|\mathcal{D}|$,
  Batch size $B$,
  Refinement network $R$ with residual
  connections.
1: Initialize random projections $A$
2: Initialize parameters $m^{(0)}, \sigma^{(0)}, C^{(0)}$
3: **for** $i = 1$ to $\mathcal{B}/\lambda$ **do**
4:   # Optimize prompt generation module
5:   Sample $\lambda$ solutions $z_i \sim m^{(t)} + \sigma^{(t)}\mathcal{N}(0, C^{(t)})$
6:   Compute the fitnesses using Equation 5
7:   Update $m^{(t)}, \sigma^{(t)}, C^{(t)}$ using the CMA-ES
8:
9:   # Optimize prediction refinement module
10:   **for** $j = 1$ to $|\mathcal{D}|/B$ **do**
11:     Sample batch $(Y_I, Y)$
12:     Compute the refined output $Y_O = Y_I + R(Y_I)$
13:     Compute the loss using Equation 6
14:     Update refinement network $R$ using AdamW
15:   **end for**
16: **end for**
17: **return** prompts $p = \mathbb{E}_z[p_0 + Az]$ and network $R$

---

divergence to constrain the output of the original black-box model and the refined output. Thus, the loss function for the CMA-ES optimizer can be formulated as follows:

$$
\mathcal{L}_I = CE(Y_I, Y) + \lambda_I * KL(Y_I \| Y_O),
\tag{5}
$$

where $\lambda_I$ is a hyper-parameter, $Y_I = f(\{[p_0 + Az, c_k]\}_{k=1}^{K}, \{i_n\}_{n=1}^{N})$ denotes the output of the black-box model, $Y$ represents the ground-truth label, $CE$ is the cross-entropy loss, and $KL$ is the Kullback-Leibler divergence. Similarly, during the optimization of the prediction refinement network, we also add a KL divergence loss, which serves as the regularization term, for training stabilization. The objective for the prediction refinement module can be written as follows:

$$
\mathcal{L}_O = CE(Y_O, Y) + \lambda_O * KL(Y_O \| Y_I),
\tag{6}
$$

where $\lambda_O$ is a hyper-parameter. Thus, the term "collaboratively" implies the algorithm can jointly optimize two modules while ensuring they work together through consistency loss rather than interfering with each other due to the sequential nature of the modules.

## 4 EXPERIMENTS

### 4.1 SETUP

**Datasets.** In accordance with CoOp (Zhou et al., 2022b), we adopt 11 distinct image classification datasets to investigate few-shot learning. These datasets encompass various domains of image classification, including generic object recognition with ImageNet (Deng et al., 2009) and Caltech101 (Li et al., 2004), fine-grained image recognition with OxfordPets (Parkhi et al., 2012), StanfordCars (Krause et al., 2013), Flowers102 (Nilsback & Zisserman, 2008), Food101 (Bossard et al., 2014) and FGVCAircraft (Maji et al., 2013), satellite image classification with EuroSAT (Helber et al., 2019), action classification with UCF101 (Soomro et al., 2012), texture classification with DTD (Cimpoi et al., 2014), and scene recognition with SUN397 (Xiao et al., 2010).

To evaluate the performance of few-shot learning models, we have followed the evaluation protocol proposed in CLIP (Radford et al., 2021). Specifically, we have trained models using 1, 2, 4, 8, and 16 shots and evaluated them on the full test sets. Additionally, we have assessed the robustness of the models to distribution shift by training CARROT on ImageNet (Deng et al., 2009) with 16 shots and evaluating it on target datasets ImageNetV2 (Recht et al., 2019), ImageNet-Sketch (Wang et al., 2019), ImageNet-A (Hendrycks et al., 2021b), and ImageNet-R (Hendrycks et al., 2021a). ImageNetV2 is a reproduced test set using different sources while following ImageNet's data collection process. ImageNet-Sketch contains sketch images belonging to the same 1,000 ImageNet classes. Both ImageNet-A and -R contains 200 classes derived from a subset of ImageNet's 1,000 classes.

**Training Details.** We utilize CLIP (Radford et al., 2021) as our black-box vision-language model, with ResNet-50 (He et al., 2016) and transformer (Vaswani et al., 2017) serving as the vision and language encoders, respectively. These encoders are initialized with CLIP's pretrained weights and kept frozen and unseen during training. To optimize the text prompts in the prompt generation module, we used the CMA-ES algorithm and set the prompt length to 4. The text prompts are projected into a subspace of dimension 512 using a random matrix sampled from a Gaussian distribution $\mathcal{N}(0, 0.02)$. The population size is set to 40, with a budget of 8,000 API calls. For the prediction refinement module, we use a three-layer MLP with a hidden dimension of 512 as the refinement network. We set the hyper-parameters $\lambda_I$ to 1 and $\lambda_O$ to 0.1 divided by the number of classes by default. The prediction refinement module is optimized using the AdamW optimizer with a learning rate of 0.001, and we set the batch size as 256 during training. Results are reported with average accuracy. All experiments are conducted on a single NVIDIA GeForce RTX 3090. We conducted three runs with different random seeds and averaged the results to obtain a reliable estimate of model performance.

**Baseline Methods.** To evaluate the effectiveness of CARROT, we compare it with three baseline methods. **(1) zero-shot CLIP:** Our first baseline method is zero-shot CLIP (Radford et al., 2021). This method requires handcrafted prompts, which we set to be the same as those used in previous works (Zhou et al., 2022b;a) to ensure a fair comparison. **(2) CoOp:** Our second baseline method is CoOp (Zhou et al., 2022b). CoOp is a white-box method that proposes learning the global text prompts through gradient descent. We use the best version of CoOp (Zhou et al., 2022b), setting the length of text prompts to 16, for comparison. **(3) BBT:** Our third baseline method is Black-Box Tuning (BBT) (Sun et al., 2022b). BBT is a black-box method for NLP tasks that proposes optimizing the soft prompts with the CMA-ES algorithm. We implement BBT in the black-box vision-language scenario, and we set its hyperparameters to be the same as those in CARROT.

## 4.2 RESULTS OF FEW-SHOT CLASSIFICATION

Figure 2 illustrates the performance of our proposed method, CARROT, in comparison to three baseline methods: CoOp (Zhou et al., 2022b), BBT (Sun et al., 2022b), and zero-shot CLIP (Radford et al., 2021), across 11 downstream datasets, accompanied by their respective average results. Our proposed approach demonstrates a significant superiority over the other black-box methods, i.e., zero-shot CLIP and BBT. Specifically, in the 16-shot setting, CARROT achieves a substantial accuracy improvement of 12.45% and 5.82% when compared to zero-shot CLIP and BBT, respectively.

Our proposed CARROT surpasses the black-box baseline BBT (Sun et al., 2022b) on most datasets except OxfordPets (Parkhi et al., 2012) and Food101 (Bossard et al., 2014). OxfordPets (Parkhi et al., 2012) and Food101 (Bossard et al., 2014) are fine-grained datasets and therefore sensitive to the fine-tuning process. Since gradient-based optimization estimates the gradient on batch inputs, it can lead to unstable training and hurt the good properties of the pretrained model. Therefore, on OxfordPets (Parkhi et al., 2012) and Food101 (Bossard et al., 2014) datasets, BBT, which is optimized without gradient, performs significantly better than gradient-based methods (CoOp (Zhou et al., 2022b) and our CARROT).

## 4.3 EFFECTIVENESS OF DIFFERENT ARCHITECTURES

We further evaluate the effectiveness of our proposed method on the 11 datasets with different visual architectures of CLIP, containing both CNNs and ViTs. Table 1 shows the results of our methods and two black-box baselines, CLIP and BBT, with different model architectures. These methods are trained on downstream 16-shot datasets.

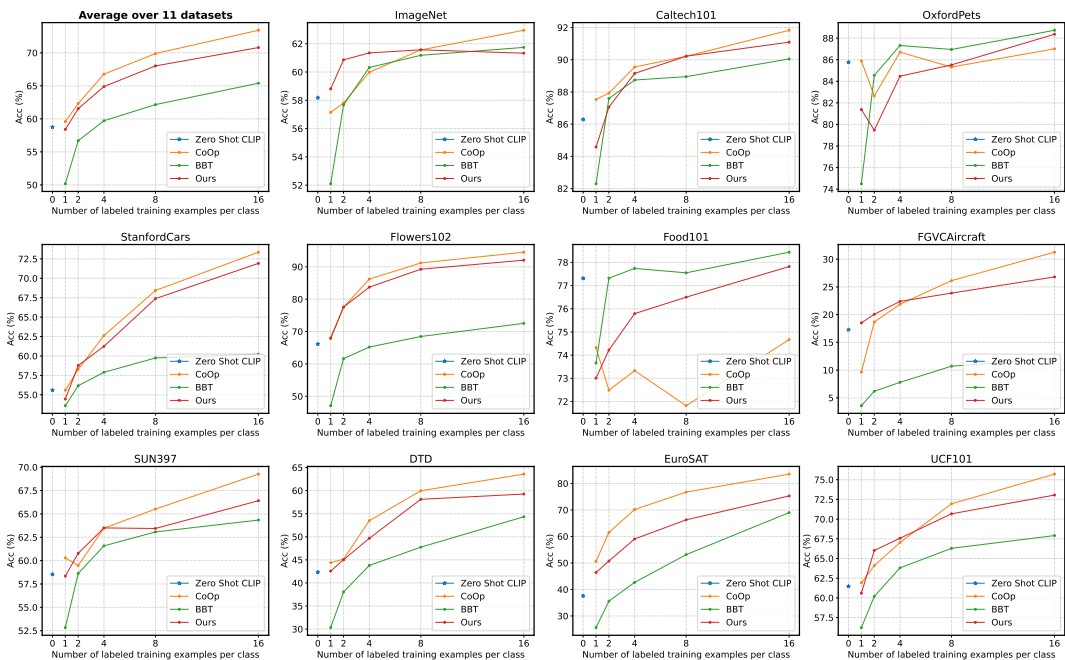

Figure 2: **Results of few-shot learning on the 11 datasets.** Here, CoOp (Zhou et al., 2022b) is a white-box method that works as the upper bound. Our proposed method greatly surpasses the black-box baseline methods, BBT (Sun et al., 2022b) and zero-shot CLIP (Radford et al., 2021).

On average, our proposed CARROT method outperformed zero-shot CLIP by 12.45%, 11.29%, 9.43%, and 10.53% on ResNet-50, ResNet-101, ViT-B/32, and ViT-B/16-based CLIP, respectively. Additionally, CARROT outperforms BBT by 5.82%, 2.82%, 2.86%, and 3.34% on average of the 11 datasets on Resnet-50, Resnet-101, ViT-B/32, and ViT-B/16 based CLIP, respectively. These results demonstrate the effectiveness of CARROT across different black-box model architectures.

Table 1: **Results of different architectures on 11 datasets.** The models are trained on the 16-shot setting datasets. **Bold** denotes the best results of black-box methods.

| Method | Pets | Flo | FGVC | DTD | EuroSAT | Cars | Food | SUN | Cal | UCF | ImageNet | Avg. |
|---|---|---|---|---|---|---|---|---|---|---|---|---|
| **ResNet-50** | | | | | | | | | | | | |
| CLIP | 85.77 | 66.14 | 17.28 | 42.32 | 37.56 | 55.61 | 77.31 | 58.52 | 86.29 | 61.46 | 58.18 | 58.77 |
| BBT | **88.73** | 72.53 | 12.07 | 54.33 | 69.01 | 60.24 | **78.44** | 64.34 | 90.05 | 67.91 | **61.74** | 65.40 |
| CARROT | 88.36 | **92.04** | **26.80** | **59.26** | **75.30** | **71.92** | 77.82 | **66.41** | **91.09** | **73.05** | 61.33 | **71.22** |
| **ResNet-101** | | | | | | | | | | | | |
| CLIP | 86.75 | 64.03 | 18.42 | 38.59 | 32.59 | 66.23 | 80.53 | 58.96 | 89.78 | 60.96 | 61.62 | 59.86 |
| BBT | **89.44** | 75.48 | 24.02 | 54.79 | 64.73 | 67.84 | **81.77** | 65.52 | 92.97 | 70.74 | **64.35** | 68.33 |
| CARROT | 89.20 | **89.61** | **28.05** | **58.85** | **67.16** | **71.57** | 81.37 | 64.79 | **93.21** | **75.16** | 63.68 | **71.15** |
| **ViT-B/32** | | | | | | | | | | | | |
| CLIP | 87.49 | 66.95 | 19.23 | 43.97 | 45.19 | 60.55 | 80.50 | 61.91 | 90.87 | 62.01 | 62.05 | 61.88 |
| BBT | **89.77** | 74.14 | 18.72 | 55.85 | 69.67 | 63.21 | **81.44** | 68.08 | **94.20** | 72.67 | **65.18** | 68.45 |
| CARROT | 88.11 | **90.43** | **26.24** | **60.40** | **70.07** | **70.45** | 77.89 | **68.57** | 93.79 | **75.72** | 62.76 | **71.31** |
| **ViT-B/16** | | | | | | | | | | | | |
| CLIP | 89.21 | 71.34 | 24.72 | 44.39 | 47.60 | 65.32 | 86.06 | 62.50 | 92.94 | 66.75 | 66.73 | 65.23 |
| BBT | **92.70** | 82.41 | 29.49 | 59.26 | 70.48 | 70.19 | **86.42** | 70.33 | **94.75** | 70.48 | **70.15** | 72.42 |
| CARROT | 91.94 | **93.92** | **36.89** | **63.28** | **72.07** | **78.11** | 83.66 | **70.97** | 94.48 | **79.78** | 68.21 | **75.76** |

## 4.4 ROBUSTNESS TO DISTRIBUTION SHIFT

We further conduct experiments to evaluate the robustness of CARROT to distribution shift. Specifically, we trained the models using the 16-shot ImageNet (Deng et al., 2009) dataset and subsequently transferred them to target domain shift datasets. These included ImageNetV2 (Recht

Table 2: **Robustness to distribution shift.** We compare our method with CLIP and CoOp (prompt length $L = 4$ and $L = 16$). And the models are trained on 16-shot datasets with different architectures. **Bold** and Underline denote the highest and second highest results.

| Method | Black-Box | Source | Target | | | | |
| --- | --- | --- | --- | --- | --- | --- | --- |
| | | ImageNet | -V2 | -Sketch | -A | -R | Avg. |
| **ResNet-50** | | | | | | | |
| CLIP | ✓ | 58.18 | 51.34 | 33.32 | 21.65 | 56.00 | 40.58 |
| CoOp (L=4) | ✗ | **63.33** | **55.40** | **34.67** | 23.06 | 56.60 | **42.43** |
| CoOp (L=16) | ✗ | 62.95 | 55.11 | 32.74 | 22.12 | 54.96 | 41.23 |
| CARROT | ✓ | 61.17 | 53.92 | 34.01 | **23.13** | **58.23** | 42.32 |
| **ResNet-101** | | | | | | | |
| CLIP | ✓ | 61.62 | 54.81 | 38.71 | 28.05 | 64.38 | 46.49 |
| CoOp (L=4) | ✗ | 65.98 | 58.60 | **40.40** | 29.60 | 64.98 | **48.40** |
| CoOp (L=16) | ✗ | **66.60** | **58.66** | 39.08 | 28.89 | 63.00 | 47.41 |
| CARROT | ✓ | 63.68 | 56.95 | 39.50 | **30.40** | **65.94** | 48.20 |
| **ViT-B/32** | | | | | | | |
| CLIP | ✓ | 62.05 | 54.79 | 40.82 | 29.57 | 65.99 | 47.79 |
| CoOp (L=4) | ✗ | 66.34 | **58.24** | **41.48** | **31.34** | 65.78 | **49.21** |
| CoOp (L=16) | ✗ | **66.85** | 58.08 | 40.44 | 30.62 | 64.45 | 48.40 |
| CARROT | ✓ | 62.76 | 56.55 | 40.26 | 31.27 | **66.46** | 48.63 |
| **ViT-B/16** | | | | | | | |
| CLIP | ✓ | 66.73 | 60.83 | 46.15 | 47.77 | 73.96 | 57.18 |
| CoOp (L=4) | ✗ | 71.73 | **64.56** | **47.89** | **49.93** | 75.14 | **59.38** |
| CoOp (L=16) | ✗ | **71.92** | 64.18 | 46.71 | 48.41 | 74.32 | 58.41 |
| CARROT | ✓ | 68.21 | 62.78 | 47.17 | 49.73 | **75.52** | 58.80 |

et al., 2019), ImageNet-Sketch (Wang et al., 2019), ImageNet-A (Hendrycks et al., 2021b), and ImageNet-R (Hendrycks et al., 2021a).

Table 2 reports the results of our method and two other baseline methods: zero-shot CLIP (Radford et al., 2021) and CoOp (Zhou et al., 2022b) (prompt length $L = 4$ and $L = 16$). Our proposed CARROT outperforms zero-shot CLIP on all datasets and architectures, with improvements of 1.74%, 1.71%, 0.84%, and 16.2% observed for ResNet-50, ResNet-101, ViT-B/32, and ViT-B/16 CLIP, respectively. These results illustrate that CARROT enhances the robustness of CLIP. Moreover, our method CARROT achieves comparable performance with the white-box prompt tuning method, CoOp. Compared with CoOp $L = 16$ variant, which performs well on few-shot classification, our CARROT achieves improvements of 1.09%, 0.79%, 0.23%, and 0.39% for each architecture, yielding the effectiveness of our method.

## 4.5 ABLATION STUDY

**Effectiveness of Components.** In this section, we analyze the efficacy of the components of CARROT. Table 4 displays the average results obtained from 11 downstream datasets for different shot settings. In the table, "PG." refers to the prompt generation module, "PR." indicates the prediction refinement module, and "Co." stands for the collaborative training algorithm.

The results demonstrate that using either the prompt generation module or the prediction refinement module in isolation achieves superior performance compared to zero-shot CLIP (58.77%) in downstream tasks. This indicates the effectiveness of both the prompt generation module and the prediction refinement module. However, when optimizing them iteratively without using the collaborative training algorithm, the model performs even worse than using the prediction refinement module alone. After incorporating the collaborative training algorithm, the models exhibit better performance compared to the other settings, indicating the effectiveness of this component. Therefore, it can be concluded that both the prompt generation module and prediction refinement module are effective, and they work best when optimized together using the collaborative training algorithm.

**Ablation of the Prediction Refinement Network.** Furthermore, we investigate the best architecture of the prediction refinement module. In Table 3, we ablate the effectiveness of the residual connection and the architecture of the refinement network $R$. As shown in Table 3, the performance of CARROT drops dramatically if we delete the residual connection (-30.91% on the 1-shot setting), indicating its effectiveness. Additionally, changing the MLP refinement network to a linear mapping also results in

Table 5: Comparison of deployment efficiency, the viability of black-box, test accuracy, training time, and memory footprint of user and server.

| Method | Black-Box | Test Accuracy | Training Time | Mem. (User) | Mem. (Server) |
|---|---|---|---|---|---|
| zero-shot CLIP (Radford et al., 2021) | ✓ | 58.18 | - | - | 244.7 MB |
| CoOp (Zhou et al., 2022b) | ✗ | 62.95 | 2h 3min | 395.7 MB | - |
| CARROT | ✓ | 61.33 | 1h 44min | 5.0 MB | 244.7 MB |

a significant performance drop. As a result, we implement the prediction refinement module using the MLP as the refinement network together with a shortcut connection.

Table 3: We ablate the components of the prediction refinement module. Arch. denotes the architecture of the refinement network. **Bold** denotes the highest result.

| | | | | shots | | |
|---|---|---|---|---|---|---|
| Residual | Arch. | 1 | 2 | 4 | 8 | 16 |
| ✗ | MLP | 28.58 | 44.44 | 52.58 | 59.25 | 63.10 |
| ✓ | Linear | 47.99 | 52.70 | 56.74 | 62.84 | 65.88 |
| ✓ | MLP | **59.49** | **61.87** | **65.26** | **68.44** | **71.22** |

Table 4: We ablate the components of CARROT. PG. denotes the prompt generation module. PR. denotes the prediction refinement module. Co. denotes the collaborative training algorithm.

| | | | | | shots | | |
|---|---|---|---|---|---|---|---|
| PG. | PR. | Co. | 1 | 2 | 4 | 8 | 16 |
| ✓ | ✗ | ✗ | 50.17 | 56.69 | 59.71 | 62.16 | 65.40 |
| ✗ | ✓ | ✗ | 54.54 | 59.07 | 63.13 | 66.07 | 69.29 |
| ✓ | ✓ | ✗ | 51.54 | 56.93 | 60.70 | 64.47 | 68.23 |
| ✓ | ✓ | ✓ | **59.49** | **61.87** | **65.26** | **68.44** | **71.22** |

**Effectiveness of Efficiency.** We further evaluate the efficiency of CARROT and compare it with CoOp (Zhou et al., 2022b) and zero-shot CLIP (Radford et al., 2021) on the ImageNet dataset, based on deployment efficiency, black-box viability, test accuracy, training time, and memory footprint. Table 5 shows that CARROT trains faster and has a significantly smaller memory footprint, using only 1/80 of the memory footprint of the white-box method CoOp (Zhou et al., 2022b), and CARROT incurs only a marginal loss in test accuracy.

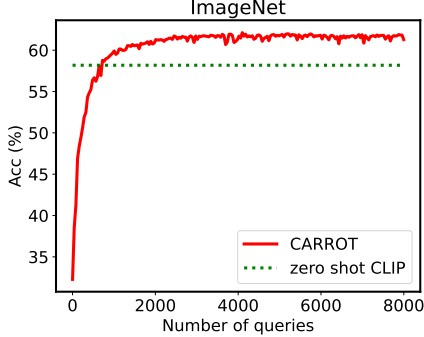

Figure 3: The relation of performance and number of queries on ImageNet.

We further report the query efficacy of our method. For the black-box vision-language model, we first upload the image dataset to the server, and we query the model with the text prompts to obtain its output $f(\{t_i\}_{i=1}^K, \{i_j\}_{j=1}^N) \in \mathbb{R}^{N \times K}$. Figure 3 depicts the relationship between the model's performance and the number of queries on the ImageNet dataset. The results indicate that our method achieves a high degree of efficacy in terms of query efficiency. Our method outperforms the zero-shot CLIP approach with a mere 1,000 queries. These results suggest its potential usefulness in a range of applications involving black-box vision-language models.

## 5 CONCLUSION

In this paper, we introduce **CollAboRative pROmpt Tuning** (**CARROT**) for fine-tuning black-box vision-language models that consists of a prompt generation module and a prediction refinement module, which are designed to learn the text prompts and refine the black-box output prediction, respectively. Additionally, we develop a novel collaborative training algorithm to optimize the modules together, even though they utilize different optimizers (derivative-free and derivative-based). We demonstrate the effectiveness of CARROT on 15 downstream datasets, as well as its robustness to distribution shifts and different architectures. Moreover, without the need for access to the parameters of vision-language models, CARROT improves its performance with marginal deployment cost and training costs. These results demonstrate the effectiveness of our method. In the future, we are going to explore the black-box fine-tuning method of VLMs in other scenarios, such as cross-dataset transfer (Zhou et al., 2022a) and unsupervised learning (Tanwisuth et al., 2023).

## REPRODUCIBILITY STATEMENT

Detailed datasets, metrics, and the implementations of our CARROT are reported in Section 4. Additionally, we have made our source code and scripts available in the supplementary materials, enabling the replication of our results and the evaluation of performance.

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
