# OpenReview forum: "Collaborative Prompt Tuning for Black-Box Vision-Language Models"
_ICLR.cc/2024/Conference — ICLR 2024 Conference Withdrawn Submission_

### Official Review · Reviewer_j3Qp · 2023-10-27

**Soundness:** 3 good
**Presentation:** 2 fair
**Contribution:** 2 fair
**Rating:** 5
**Confidence:** 4

**Summary:**

The article introduces a prompt-tuning framework for black-box visual language models. In order to fine-tune the black-box model, the authors propose building two modules. The first module maps low-dimensional vectors to high-dimensional matrices and adjusts the prompt by adding it to the mapped matrix. The second module modifies the logit output of the black-box model using a residual network. For the first module, the authors use the CMA-ES method, which does not require derivatives, for optimization. For the second module, the AdamW method is used for optimization. To address training issues caused by differences in optimization methods, the authors further propose a joint optimization approach. The training process is splitted to several rounds and in each round, the optimization of the second module starts after the first has been optimized. They also introduce KL divergence in the optimization objectives of each module to stabilize the optimization process. The authors conducted few-shot experiments on multiple datasets. The experimental results demonstrate that their method outperforms three baseline methods: zero-shot CLIP, CoOp and BBT, and they also require much less memory for deployment.

**Strengths:**

originality: The problem addressed in the article is highly relevant and innovative. It is almost the pioneering work in exploring efficient fine-tuning methods for black-box vision-language models.

quality: The article demonstrates high quality as it conducts extensive experiments on multiple datasets. Abundant ablation experiments also validate the roles of various modules in the framework.

clarity: The description of formulas and methods is fairly clear, although there are some issues such as missing variables that will be pointed out in the questions.

significance: The proposed method holds some significance for PEFT, particularly in terms of its memory efficiency.

**Weaknesses:**

originality: The prompt generation module in the proposed framework bears significant similarity to the method mentioned in the paper "Black-Box Tuning for Language-Model-as-a-Service". The article also fails to discuss the differences between their approach and BBT in the related work section. Nonetheless, the other modules exhibit some degree of novelty.

clarity: One major concern is how the black-box is defined exactly. The method described in the paper requires modifying the embedding of the prompt and accessing the model's returned logits, which seems inconsistent with the common definition of a black-box VLM.

significance: In reality, very few companies providing VLM services openly disclose their embedding models or return logits. Therefore, the practical value of the article's approach for tuning black-box VLMs is questionable.

**Questions:**

1. The core question is why this paper can access the embedding of the prompt and the final returned logits in a black-box setting, which does not match with the reality of model services.

2. In the experimental section, the paper mentions conducting experiments with three different random seeds and averaging the results. Could the authors confirm whether the random seeds are fixed for each method?

3. Why were these 11 datasets chosen for testing? What was the rationale behind their selection?

4. Formula 2 does not include λ, but it is mentioned in the following explanation. Please check it.

5. In Section 3.1, the sentence "Next, we add the prompts to the initial prompts p0 (e.g., 'a photo of a')." refers to the prompt in textual form, but the preceding paragraph states that the prompt should be a matrix. It is suggested to improve the consistency in defining the prompt throughout the paper.

6. In Section 3.3, the paper states, "Fortunately, vision-language models typically have residual connections, and the prediction refinement network also comprises a shortcut connection. Therefore, we can iteratively optimize the prompt generation module and the prediction refinement module." The reviewer would like to know why the presence of residual structures in the original VLM facilitates collaborative training. Could you provide some explanations or references supporting this viewpoint?

---

### Official Review · Reviewer_Nv8C · 2023-10-31

**Soundness:** 3 good
**Presentation:** 3 good
**Contribution:** 2 fair
**Rating:** 5
**Confidence:** 4

**Summary:**

This paper studies the black box optimization problem for pre-trained VL models. With the increasing cost of training, model owners tend to protect their digital assets, making existing fine-tuning methods inapplicable. To address this, the paper proposes a method called CARROT, which allows fine-tuning models without accessing the model parameters, enabling adaptation to downstream tasks. CARROT consists of a prompt generation module and a prediction refinement module. The prompt generation module learns the text prompt, while the prediction refinement module enhances the model's prediction results. Due to the black box nature, CARROT cannot be optimized end-to-end like traditional fine-tuning. Therefore, the paper proposes using CMA-ES and AdamW optimizer to separately optimize the two modules and introduces KL loss as a constraint to ensure consistency between the outputs of the two modules. The paper conducts experiments on the CoOp benchmark, and the results demonstrate a significant improvement over previous methods.

**Strengths:**

The black box problem addressed in this paper is important.

**Weaknesses:**

1. This paper lacks a comparison with related methods, such as [1]. As both [1] and this paper apply the CMA-ES algorithm, it is important to clarify their differences. The results of [1] were publicly available on GitHub in April, and it seems that this paper does not have significant advantages over [1] in terms of performance.

2. The ablation experiments are insufficient. For example, in eq (5) and (6), the impact of introducing KL divergence should be demonstrated through ablation experiments. Additionally, the value of $\lambda$ should be analyzed through experiments.

3. The authors should validate the generalization and versatility of their method on more VL-pretrained models and tasks. For example, they can consider using EVA-CLIP as the base model and include tasks such as text-image retrieval and open-vocabulary segmentation. The current validation only focuses on the CoOp benchmark, which is not sufficient to demonstrate the superiority of their method.

[1.] Yu, Lang, et al. "Black-box Prompt Tuning for Vision-Language Model as a Service."

**Questions:**

No other questions.

---

### Official Review · Reviewer_2nQv · 2023-10-31

**Soundness:** 3 good
**Presentation:** 3 good
**Contribution:** 2 fair
**Rating:** 5
**Confidence:** 2

**Summary:**

The paper proposed a new method for fine-tuning black-box vision-language models (VLMs) without accessing their parameters. The method is called CARROT and it has three main components:

-A prompt generation module that learns text prompts for the input of the black-box VLM using derivative-free optimization.

-A prediction refinement module that improves the output of the black-box VLM using a residual network and gradient descent.

-A collaborative training algorithm that alternates between optimizing the two modules and uses a prediction-consistent loss to ensure stability.

The paper claims that CARROT can achieve significant improvements over zero-shot learning and black-box baselines on 15 few-shot classification datasets. It also compares favorably to the white-box method in terms of training speed and memory footprint.

**Strengths:**

The paper presents a novel method called CARROT for fine-tuning black-box vision-language models (VLMs) without accessing their parameters. The method consists of three main components: a prompt generation module, a prediction refinement module, and a collaborative training algorithm. The paper claims that CARROT can achieve significant improvements over zero-shot learning and black-box baselines on 15 few-shot classification datasets. It also compares favorably to the white-box method in terms of training speed and memory footprint.

One of the strengths of the paper is that it addresses an important problem in the field of machine learning, namely how to fine-tune black-box models without access to their parameters. The authors provide a detailed description of the CARROT method and its three main components, which makes it easy to understand and replicate. They also provide extensive experimental results that demonstrate the effectiveness of the method on a variety of few-shot classification tasks.

Another strength of the paper is that it compares CARROT to both zero-shot learning and black-box baselines, as well as to the white-box method. This allows readers to see how CARROT performs relative to other methods and provides a more complete picture of its strengths and weaknesses. The authors also provide a detailed analysis of the results, which helps readers understand why CARROT performs better than other methods.

Overall, the paper makes a contribution to the field of machine learning by presenting a new method for fine-tuning black-box VLMs that is both effective and efficient. The authors provide extensive experimental results that demonstrate the effectiveness of the method on a variety of few-shot classification tasks, which should be useful for researchers working in this area.

**Weaknesses:**

(1) The proposed CARROT algorithm is composed of three parts, i.e., the black-box PG, the white-box PR, and the Co training paradigm. Since PG is directly borrowed from [1] and PR is not black-box, I believe that Co is the most important contribution in this paper. However, as shown in Table 4, why PR and PG conflict with each other, and why Co resolves such conflict are unclear to me. This is the core part of CARROT. More theoretical or empirical explanations are strongly needed.

(2) I noticed that there is a concurrent paper [2] in this topic (already open-source), which also applies evolutionary algorithms (including CMA-ES) to conduct prompt learning but in either a shallow or a deep manner. Since this paper has no other black-box methods to compare with (BBT in Table 1 and Figure 2 is just PG, a part of CARROT), I strongly suggest the authors make comparisons with [2] in fair experimental settings.

(3) Ablation studies on the subspace dimension in CMA-ES, \lambda_I, and \lambda_O are all missing.

Minor flaws
(4) The font sizes in Figure 1 are inconsistent, and the icon of Co is kind of confusing to me.

(5) The ImageNet acc is 61.33% in Table 1 but becomes 61.17 in Table 2. Please check the reported results.

[1] ICML 2022. Black-Box Tuning for Language-Model-as-a-Service

[2] IJCAI 2023. Black-box Prompt Tuning for Vision-Language Model as a Service

**Questions:**

Please reply to the five weakness questions during the rebuttal.

---

### Official Review · Reviewer_S6pY · 2023-11-01

**Soundness:** 3 good
**Presentation:** 3 good
**Contribution:** 2 fair
**Rating:** 5
**Confidence:** 3

**Summary:**

This paper proposes a Prompt Tuning method for black-box Vision-Language model (VLM), which designs a better text prompt and refines the output of the VLM. According to the experiments, the few-shot performance of VLM is improved on the classification task, and the proposed collaborative training approach integrates the derivative-free and derivative-based model optimization well.

**Strengths:**

1.	The overall writing of the paper is clear, and the tables and figures intuitively designed.
2.	From the experiments, the proposed CARROT method achieves good improvement on the few-shot classification task.

**Weaknesses:**

1.	The paper seems not novel, the proposed Prompt Generation Module is just a direct transfer of the language model's Black-Box Prompt Tuning method [1] to VLM directly, and there is no specific design of the image modality in VLM, which makes the method very incremental.
2.	There has been work [2] on Black-Box's VLM Prompt Tuning and it is specifically designed for both text and image modalities. It also uses CMA-ES optimization and transfers the data to low-dimensional subspace. While this work is not mentioned in this paper, could you please elaborate on the differences between the proposed approach and this paper and clarify your contribution.
3.	Sec 3.3 in the article is puzzling, especially the second paragraph, it is hard to understand why the residual connection is causally related to iteratively optimizing two modules?
4.	Why include λI ∗ KL(YI ∥ YO) in Eq. 5, and what is the purpose of the Refinement module if one wants YI to converge to YO? It seems more reasonable to keep only the KL divergence calculation in Eq. 6. Please prove the design of this loss function by ablation experiments.
5.	The methods compared in the article are too few, please compare with some sota PT [3,4] methods to further prove the validity of the methodology.
Reference
[1] Sun, Tianxiang, et al. "Black-box tuning for language-model-as-a-service." International Conference on Machine Learning. PMLR, 2022.
[2] Yu, Lang, et al. "Black-box Prompt Tuning for Vision-Language Model as a Service." IJCAI, 2023.
[3] Zhou, Kaiyang, et al. "Conditional prompt learning for vision-language models." Proceedings of the IEEE/CVF Conference on Computer Vision and Pattern Recognition. 2022.
[4] Sun, Tianxiang, et al. "BBTv2: towards a gradient-free future with large language models." Proceedings of the 2022 Conference on Empirical Methods in Natural Language Processing. 2022.

**Questions:**

Please refer to the weaknesses